# Familial Gastrointestinal Stromal Tumor Associated with Zebra-like Pigmentation

**DOI:** 10.3390/biomedicines11061590

**Published:** 2023-05-30

**Authors:** Takuma Hayashi, Ikuo Konishi

**Affiliations:** 1Cancer Medicine, National Hospital Organization Kyoto Medical Center, Kyoto 612-0861, Japan; 2First-Track Medical R&D, The Japan Agency for Medical Research and Development (AMED), Tokyo 100-0004, Japan; 3Department of Obstetrics and Gynecology, Kyoto University School of Medicine, Kyoto 606-8303, Japan

**Keywords:** KIT, GIST, cancer genetic test, imatinib, pathogenic variant, tyrosine kinase inhibitor

## Abstract

**Purpose:** According to clinical studies, gastrointestinal stromal tumors (GISTs) are predominantly sporadic. GISTs associated with familial syndromes are very rare, and most patients exhibit wild-type KIT and platelet-derived growth factor alpha (PDGFRA). To date, GISTs associated with germline *KIT* pathogenic variants have been observed in only 30 kindreds worldwide. The efficacy of imatinib, a multityrosine kinase inhibitor, in patients with GIST presenting germline *KIT* variants has been poorly reported, and the efficacy in clinical trials of treatments with tyrosine kinase inhibitors remains unclear. Therefore, imatinib is not yet recommended for treating GIST patients with germline *KIT* variants. **Experimental Design:** We performed cancer genomic testing on samples from a 32-year-old male patient with advanced GISTs throughout the upper stomach and cutaneous hyperpigmentation to determine diagnosis and treatment strategies. **Results:** We detected a germline W557R pathogenic variant of *KIT*. The patient was diagnosed with familial multinodular GIST based on the clinical findings and familial history of malignant tumors. Treatment with imatinib resulted in long-term regression of GISTs. **Conclusions:** Pathogenic variants detected by cancer genome testing can be used to diagnose malignant tumors and select new therapeutic agents for patients with advanced malignancies.

## 1. Introduction

More than 90% of gastric cancer cases are adenocarcinomas arising from the mucosal epithelial cells of the stomach wall. Adenocarcinomas are classified into two types, differentiated and undifferentiated gastric cancers, depending on their growing patterns [1]. In differentiated gastric cancer, cancer cells proliferate while forming glandular ductal structures, whereas cancer cells proliferate and scatter in undifferentiated gastric cancer. Undifferentiated gastric cancer is considered more malignant than differentiated gastric cancer. Undifferentiated gastric cancer includes scirrhous gastric cancer, which grows rapidly. Since Gastrointestinal Stromal Tumors (GIST) form mass-like lesions under the mucosal tissue, GISTs exhibit properties different from those of gastric and colorectal cancers, which develop from mucosal tissue [2]. In many cases, GISTs occur under the mucosal tissue of the stomach and small intestine, and the onset of GISTs in tissues of the large intestine and esophagus is rare.

On the other hand, hereditary gastric cancer (i.e., familial gastric cancer), which is due to mutations in specific genes, has been described. An autosomal dominant mode of inheritance causes Hereditary Diffuse Gastric Cancer (HDGC). Diffuse gastric cancer is also referred to as signet-ring cell carcinoma or isolated cell-type carcinoma. HDGC is characterized by tumors with the morphology of a poorly differentiated adenocarcinoma in which wall thickening occurs because the tumor cells constituting the HDGC invade the gastric wall without forming a clear mass [3]. HDGC is a rare malignancy characterized by autosomal dominant inheritance of pathological variants of the *cadherin 1 (CDH1)* gene encoding E-cadherin, which is involved in cell-to-cell adhesion, maintenance of epithelial architecture, tumor suppression, and regulation of intracellular signaling pathways. Germline mutation in the tumor suppressor gene *CDH1* is also associated with lobular breast cancer (LBC). The median age of onset of HDGC is 38 years (range: 14–69 years), and many cases with *CDH1* pathogenic variants present with HDGC onset before age 40 years [4]. The estimated cumulative risk of HDGC by age 80 is 70% in men and 56% in women [4]. In addition, the risk of developing LBC for women with HDGC has been reported to be 42% [5].

GISTs are intestinal mesenchymal tumors that arise from pacemaker cells called Cajal cells, which play a role in gastric motility [6]. *KIT* proto-oncogene (*KIT*), also known as cluster of differentiation (CD)117, is important for the development of Cajal interstitial cells, mast cells, and melanocytes [7]. Familial GISTs are a rare autosomal dominant disorder, which affects only a few families worldwide. It is mostly caused by germinal *KIT* pathogenic variants, which are present in approximately 75–80% of cases [8]. The KIT ligand (KITLG)/c-kit and KRAS proto-oncogene (KRAS)/mitogen-activated protein kinase (MAPK) pathways have been identified as important pathways for controlling pigmentation [9]. At our general medical institution, a 32-year-old man underwent a pyloric side gastrectomy for a tumor found in the body of the stomach. As a result of surgical pathological examinations, he was diagnosed with Stage IIa undifferentiated adenocarcinoma. In the next year, contrast-enhanced CT imaging showed a submucosal tumor in the upper stomach, a tumor in the liver, and multiple tumors in the abdominal cavity. Pigmentation was observed on the skin throughout the body. A cancer genetic examination was performed on a tumor in the upper stomach, and was resected by a gastroscope; the genetic test revealed a *KIT* pathogenic variant (W557R). Therefore, we report a case of familial GISTs with a novel germline pathogenic variant within exon 11 of the *KIT* gene, which was detected using cancer genome testing. The young adult male patient presented with gastric subepithelial lesions accompanied by skin hyperpigmentation and was subsequently diagnosed with multinodular GISTs based on the clinical findings and familial history of malignant tumors. Administration of imatinib, a tyrosine kinase inhibitor, has been identified as an effective targeted therapy against GISTs with *KIT* pathogenic variants, i.e., druggable variants [8]. Thus, the patient is currently treated with imatinib.

## 2. Materials and Methods

### 2.1. Magnetic Resonance Imaging (MRI) Examination

To determine the presence, size, and location of the patient’s mass, the contrast-enhanced MRI was performed to localize the patient’s mass sing MRI equipment (Vantage Centurian: Vantage Galan 3T MRT-3020, Canon Medical Systems, Inc., Ohtawara, Tochigi, Japan).

### 2.2. Laparoscopic Surgery

A pyloric gastrectomy and reconstruction of the remnant stomach and duodenum were performed by laparoscopic surgery using a laparoscope (ENDOEYE FLEX 3D, Olympus Corporation, Shinjuku, Tokyo, Japan) and a surgical device (HICURA, Olympus Corporation, Shinjuku, Tokyo, Japan) to surgically treat the gastric corpus area where the gastric tumor was found.

### 2.3. Cancer Genomic Testing

In October 2022, a patient had mild hepatic dysfunction after resection of the upper part of the stomach for surgical treatment. Contrast-enhanced computed tomography (CT) imaging showed a submucosal mass in the body of the stomach and a disseminated metastatic mass in the abdominal cavity. In December 2022, our medical staff resected the tumor site in the stomach using laparoscopic surgery. We performed cancer genomic testing (FoundationOne^®^ CDx’s cancer genome test, Foundation Medicine, Inc., Cambridge, MA, USA) using tissue sections of the resected tumor to determine the diagnosis and treatment strategies.

### 2.4. Histopathological Examination

To assess the gross and histopathological characteristics of the resected specimens, a surgical pathologist performed a histopathological analysis of sections from formalin-fixed paraffin-embedded resected tissue.

Hematoxylin and eosin staining and immunohistochemistry analyses were performed using anti-human a-smooth muscle actin (a-SMA) antibody (#NBP2-44464, Novus Biologicals, LLC, Centennial, CO, USA), anti-human CD34 antibody (#NBP2-44570, Novus Biologicals, LLC), anti-human c-KIT antibody (#NBP2-29423, Novus Biologicals, LLC), anti-human desmin antibody (#NB120-17156, Novus Biologicals, LLC), and anti-human deleted-in-oral-cancer-1 (DOC1) antibody (#LS-C669557-0.1, LifeSpan BioSciences, Inc., Shirley, MA, USA) following standard procedures.

## 3. Results

### 3.1. Histopathology and Contrast-Enhanced CT Imaging Analyses

A 32-year-old male patient underwent pyloric gastrectomy for a gastric tumor. A reconstruction was performed by suturing the remnant stomach on the cardiac side and the duodenum (Billroth I method). Pathological analysis of the resected tissue revealed spindle-shaped atypical cells that were intricately arranged in an east-like manner (Figure 1A). Mitotic counts were <5 per 50 high-power fields (Figure 1A). Recent clinical research has revealed that tissue sections of GISTs are positively stained for KIT (CD117, 95%) and DOC1 (almost exclusively associated with GIST) [8]. The resected tissue sections were not immunostained for desumin, α-SMA, and CD34, whereas they expressed c-KIT (CD117) and DOC1 (Figure 1A). The patient was diagnosed with stage IIa undifferentiated adenocarcinoma. In the following year, the patient presented with chief complaints of abdominal pain and anorexia. Contrast-enhanced CT revealed a submucosal tumor in the upper stomach, a tumor in the liver, and multiple tumors in the abdominal cavity (Figure 1B).

### 3.2. Diagnosis by Cancer Genetic Test

A recent report revealed that a de novo genetic variant in the *KIT* (Signaling from the tyrosine kinase receptor, KIT is essential for primordial germ cell growth both in vivo and in vitro. Many downstream effectors of the KIT signaling pathway have been identified in other cell types, but how these molecules control primordial germ cell survival and proliferation are unknown. Determination of the KIT effectors acting in primordial germ cells has been hampered by the lack of effective methods to easily manipulate gene expression in these cells. In addition to its role in hematopoietic maintenance, growth, and differentiation, the KIT signaling pathway regulates cell shape, motility, and adhesion via cytoskeletal changes) gene causes atypical lentiginosis and hyperpigmentation in pediatric patients or young adults [9]. We observed a zebra-like dark pigmentation on the patient’s body (Figure 2). A cancer genetic test (FoundationOne^®^ CDx tissue, Foundation Medicine, Inc., Cambridge, MA, USA) was performed on samples of the upper stomach tumor resected using a gastroscope. It revealed a pathogenic variant (W557R) located within exon 11 of the *KIT* gene (allele-fraction = 0.5798). The results were verified using the ClinVar (The National Center for Biotechnology Information (NCBI) advances science and health by providing access to biomedical and genomic information. ClinVar is a database provided by NCBI as a freely available archive that collects information about human genomic diversity and associated diseases) human genome database (Appendix A). The detected W557R *KIT* pathogenic variant may result from a germinal mutation based on the allele-fraction value obtained from the cancer genetic testing. The patient was diagnosed with familial multinodular GISTs based on the clinical findings and familial history of malignant tumors (Appendix A). The pathogenic variant encodes an activated form of KIT that promotes the synthesis of melanin pigment. Consistently, we observed pigmentation in the patient (Appendix A) [10].

Tumor cell models have been developed for rapid screening of candidate drugs and investigation of malignant tumor mechanisms. The receptor tyrosine kinase KIT is involved in intracellular signaling, and a mutated form of KIT plays an important role in the development of some malignant tumors. KIT independently activates three pathways, namely phosphatidylinositol-3 kinase (PI3K)/AKT (protein kinase B), KRAS/MAPK/extracellular signal-regulated kinase (ERK), and Janus kinase (JAK)/signal transducer and activator of transcription (STAT) pathways (Appendix A) [11]. The PI3K pathway activates antiapoptotic genes, thus promoting cell survival. The MAPK/ERK pathway regulates genes involved in cell proliferation. The JAK/STAT pathway is also associated with cell proliferation [11]. Therefore, *KIT* pathogenic variants significantly activate the PI3K/AKT, MAPK/ERK, and JAK/STAT pathways, which can be inhibited by imatinib, a tyrosine kinase inhibitor (KIT (also called c-Kit), a receptor tyrosine kinase, is involved in intracellular signaling pathways, and the KIT with pathogenic variants plays a crucial role in the development of some malignant tumors. The biological function of KIT has led to the concept that inhibiting tyrosine kinase activity can be a critical target for clinical cancer therapy. Several small molecules or antibody drugs, which are identified as tyrosine kinase inhibitors (TKIs), markedly inhibit the KIT (c-Kit) signaling pathway, an effect associated with malignant tumor suppression and depigmentation) [11]. Because imatinib administration might be an effective targeted therapy against GISTs associated with *KIT* pathogenic variants, the patient is currently treated with imatinib. Moreover, a clear histopathological reaction was observed after surgical treatment.

## 4. Discussion

A 32-year-old man underwent a pyloric side gastrectomy for a tumor found in the body of the stomach. Pathological diagnosis using the resected tissue revealed that spindle-shaped atypical cells were intricately arranged in an east-like manner. As a result of these surgical pathological examinations, he was diagnosed with Stage IIa undifferentiated adenocarcinoma. In the next year, he presented with chief complaints of abdominal pain and anorexia. Contrast-enhanced CT imaging showed a submucosal tumor in the upper stomach, a tumor in the liver, and multiple tumors in the abdominal cavity. Pigmentation was observed on the skin throughout the body. A cancer genetic examination was performed with a tumor in the upper stomach resected by a gastroscope, and the genetic test revealed a *KIT* pathogenic variant (W557R). Diagnosis of familial GIST was made based on his clinical findings and family history of cancer. Recent clinical studies demonstrated that the tyrosine kinase inhibitor imatinib significantly improved the survival of patients with malignancies expressing KIT pathogenic variants [12]. Clinical treatment with imatinib has been associated with long-term regression of GISTs. Now, he has been on treatment with imatinib, a tyrosine kinase inhibitor.

Reports from clinical studies have revealed that GISTs are predominantly sporadic. The occurrence of GISTs associated with familial syndromes is extremely rare, and most cases present wild-type KIT and platelet-derived growth factor alpha (PDGFRA) [13]. To date, GISTs with germline KIT pathogenic variants have been described in only 30 families worldwide [14]. Here, we report the case of a 32-year-old male patient diagnosed in the cancer genomic medicine unit with advanced GISTs throughout the upper stomach and hyperpigmented skin. The patient exhibited a germline W557R pathogenic variant of *KIT*. Treatment with imatinib, selected from the results of cancer genomic testing, resulted in long-term regression of GISTs with clear pathological responses after surgical treatment.

Cancer genomic medicine examines genetic changes occurring in the cancer cells of patients and provides treatments that are tailored to the characteristics of each patient with malignancies. In Japan, health insurances cover the costs of cancer gene panel examination since June 2019, and cancer patients can benefit from cancer genomic medicine [15,16]. Cancer gene panel examination simultaneously investigates multiple genes in cells from resected cancer tissue. If pathogenic variants (i.e., druggable variants) are detected, drugs or substances tested in clinical trials/clinical studies potentially effective against the pathogenic variants are selected, including from the cancer genomic database, based on clinical guidelines on chemotherapy [14,15]. In other words, when a cancer genome test detects pathogenic variants of genes in tumor-constituting cells, it is preferable to select antitumor agents that specifically act on the pathogenic variants.

In clinical practice, pathogenic variants (i.e., druggable variants) detected by cancer genome testing can be used to diagnose malignant tumors and select new therapeutic agents for many patients with advanced malignancies. To the best of our knowledge, this is the first report of the germinal *KIT* W557R pathogenic variant in familial GISTs. The present finding provides additional understanding of the disease pathogenesis and valuable information for precision treatment. At present, the number of patients targeted by cancer genomic medicine is very small, and the percentage of patients for whom a potential treatment was identified is also low. Based on these facts, accurate explanations to patients are required for promoting the use of cancer genomic medicine. Furthermore, future progresses in research and development including clinical trials are awaited.

## 5. Conclusions

Advances in science and technology have been making remarkable progress in medicine. In clinical practice, pathogenic variants detected by cancer genome testing can be used to diagnose malignant tumors and select new therapeutic agents for patients with advanced malignancies.

## Figures and Tables

**Figure 1 biomedicines-11-01590-f001:**
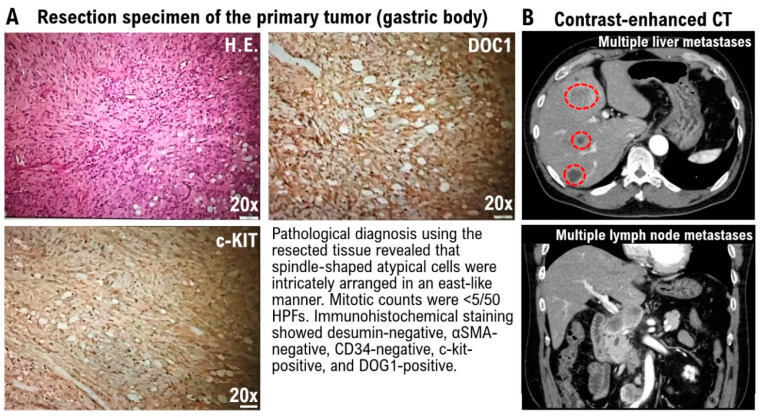
Pathological and contrast-enhanced computed tomography (CT) analyses. (**A**) Results of the pathological examination. Magnification: 20×. Scale bar = 100 μm. (**B**). Contrast-enhanced CT revealed a submucosal tumor in the upper stomach, a tumor in the liver, and multiple tumors in the abdominal cavity. Liver metastases of gastrointestinal stromal tumors (GISTs) are indicated by red dotted lines.

**Figure 2 biomedicines-11-01590-f002:**
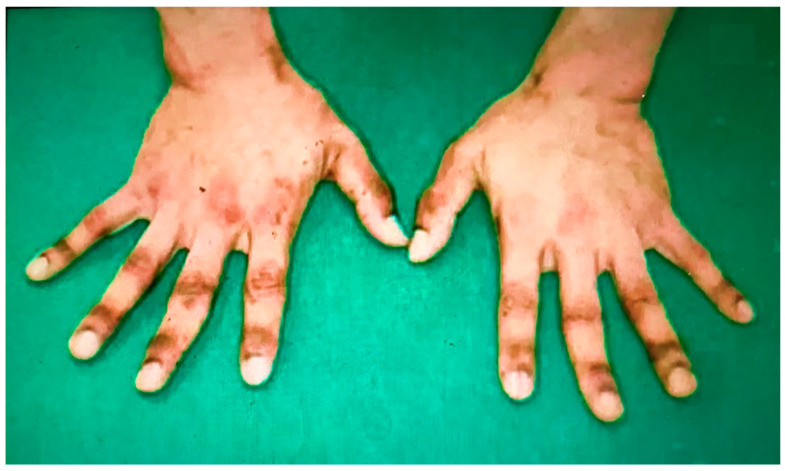
Picture of the patient’s hands before treatment with imatinib mesylate. The back of both hands exhibited zebra-like pigmentation.

## Data Availability

Data available on request due to restrictions eg privacy or ethical The data presented in this study are available on request from the corresponding author.

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
