# Peer review of "Familial Gastrointestinal Stromal Tumor Associated with Zebra-like Pigmentation"

_biomedicines, 2023, doi:10.3390/biomedicines11061590_

Round 1
Reviewer 1 Report
The article is devoted to an actual topic. GIST tumors remain a significant problem associated with the presence of diagnostic contradictions. A clinical case is presented that shows the use of molecular markers as parameters of differential diagnosis. However, it is worth noting the ambiguity of the introduction and discussion. It is necessary to rewrite them, indicating the practical component of the work.
The article is devoted to an actual topic. GIST tumors remain a significant problem associated with the presence of diagnostic contradictions. A clinical case is presented that shows the use of molecular markers as parameters of differential diagnosis. However, it is worth noting the ambiguity of the introduction and discussion. It is necessary to rewrite them, indicating the practical component of the work.
Author Response
Manuscript ID: biomedicines-2426645
Reviewer 1.
Comments and Suggestions for Authors
Comment 1. The article is devoted to an actual topic. GIST tumors remain a significant problem associated with the presence of diagnostic contradictions. A clinical case is presented that shows the use of molecular markers as parameters of differential diagnosis. However, it is worth noting the ambiguity of the introduction and discussion. It is necessary to rewrite them, indicating the practical component of the work.
Answer 1. We appreciate your comment. Following the reviewer's comments, we have rewritten the introduction and the discussion as followings.
Section of Introduction
GISTs are intestinal mesenchymal tumors that arise from pacemaker cells called Cajal cells, which play a role in gastric motility.6 KIT proto-oncogene (KIT), also known as cluster of differentiation (CD)117, is important for the development of Cajal interstitial cells, mast cells, and melanocytes.7 Familial GISTs are a rare autosomal dominant disorder, which affects only a few families worldwide. It is mostly caused by germinal KIT pathogenic variants, which are present in approximately 75%–80% of cases.8 The KIT ligand (KITLG)/c-kit and KRAS proto-oncogene (KRAS)/mitogen-activated protein kinase (MAPK) pathways have been identified as important pathways for controlling pigmentation.9 At our general medical institution, a 32-year-old man underwent a pyloric side gastrectomy for a tumor found in the body of the stomach. As a result of surgical pathological examinations, he was diagnosed with Stage IIa undifferentiated adenocarcinoma. In next year, Contrast-enhanced CT imaging showed a submucosal tumor in the upper stomach, a tumor in the liver, and multiple tumors in the abdominal cavity. Pigmentation was observed on the skin throughout the body. A cancer genetic examination was performed with a tumor in the upper stomach resected by gastroscope, and the genetic test revealed KIT pathogenic variant (W557R). Therefor, here, we report a case of familial GISTs with a novel germline pathogenic variant within exon 11 of the KIT gene, which was detected using cancer genome testing. The young adult male patient presented with gastric subepithelial lesions accompanied by skin hyperpigmentation and was subsequently diagnosed with multinodular GISTs based on the clinical findings and familial history of malignant tumors. Administration of imatinib, a tyrosine kinase inhibitor, has been identified as an effective targeted therapy against GISTs with KIT pathogenic variants, i.e., druggable variants.8 Thus, the patient is currently treated with imatinib.
Section of Discussion
- Discussion
Our cancer genomic medicine unit used cancer genomic testing (FoundationOne® CDx tissue) and identified a germline KIT W557R pathogenic variant in a 32-year-old male patient with advanced GISTs throughout the upper stomach and cutaneous hyperpigmentation. A 32-year-old man underwent a pyloric side gastrectomy for a tumor found in the body of the stomach. Pathological diagnosis using the resected tissue revealed that spindle-shaped atypical cells were intricately arranged in an east-like manner. As a result of these surgical pathological examinations, he was diagnosed with Stage IIa undifferentiated adenocarcinoma. In next year, he presented with chief complaints of abdominal pain and anorexia. Contrast-enhanced CT imaging showed a submucosal tumor in the upper stomach, a tumor in the liver, and multiple tumors in the abdominal cavity. Pigmentation was observed on the skin throughout the body. A cancer genetic examination was performed with a tumor in the upper stomach resected by gastroscope, and the genetic test revealed KIT pathogenic variant (W557R). Diagnosis of familial GIST was made based on his clinical findings and family history of cancer. Recent clinical studies demonstrated that the tyrosine kinase inhibitor imatinib significantly improved survival of patients with malignancies expressing KIT pathogenic variants.12 Clinical treatment with imatinib has been associated with long-term regression of GISTs. Now, he has been on treatment with imatinib, a tyrosine kinase inhibitor.

Reviewer 2 Report
This is an interesting case report of a familial gastrointestinal stromal tumor associated to a zebra-like pigmentation.
The case is well-described in each part by the authors.
The information provided could be useful for the readers of this journal.
I would suggest only to add more keywords: only three keywords are currently reported.
Author Response
Manuscript ID: biomedicines-2426645
Reviewer 2.
Comments and Suggestions for Authors
Comment 1. This is an interesting case report of a familial gastrointestinal stromal tumor associated to a zebra-like pigmentation.
Answer 1. We appreciate your comment.
Comment 2. The case is well-described in each part by the authors.
Answer 2. We appreciate your comment.
Comment 3. The information provided could be useful for the readers of this journal.
Answer 3. We appreciate your comment.
Comment 4. I would suggest only to add more keywords: only three keywords are currently reported.
Answer 4. We appreciate your comment. Following the reviewer's comments, we have added three more keywords to the revised manuscript as following.
KIT; GIST; cancer genetic test; imatinib; pathogenic variant, tyrosine kinase inhibitor
